Impact of concurrent aerobic and resistance training on body composition, lipid metabolism and physical function in patients with type 2 diabetes and overweight/obesity: a systematic review and meta-analysis

AL-Mhanna Sameer Badri 1 2
Alghannam Abdullah F. 3
http://orcid.org/0009-0009-4749-3248 Alkhamees Nouf H. 4
Sheeha Bodor Bin 4
Omar Norsuhana 2 suhanakk@usm.my
Albalawi Hani 5
http://orcid.org/0000-0001-7633-7900 Gülü Mehmet 6
Canli Umut 7
Afolabi Hafeez Abiola 8
http://orcid.org/0000-0002-7299-1072 Abubakar Bishir Daku 9
http://orcid.org/0000-0003-4100-8765 Badicu Georgian 10
Ahmad Rozaziana 2
Grivas Gerasimos V. 11
http://orcid.org/0000-0003-0844-1284 Batrakoulis Alexios 12 13
1 Center for Global Health Research, Saveetha Medical College and Hospitals, Saveetha Institute of Medical and Technical Sciences, Saveetha University , Chennai , India
2 Department of Physiology, School of Medical Sciences, Universiti Sains Malaysia, Kubang Kerian , Kelantan , Malaysia
3 Lifestyle and Health Research Center, Health Sciences Research Center, Princess Nourah bint Abdulrahman University , Riyadh , Saudi Arabia
4 Department of Rehabilitation Sciences, College of Health and Rehabilitation Sciences, Princess Nourah bint Abdulrahman University , Riyadh , Saudi Arabia
5 Department of Health Rehabilitation Sciences, Faculty of Applied Medical Sciences, University of Tabuk , Tabuk , Saudi Arabia
6 Department of Sports Management, Faculty of Sport Sciences, Kirikkale University , Kirikkale , Turkey
7 Sports Science Faculty, Tekirdag Namik Kemal University , Tekirdag , Turkey
8 Department of General Surgery, School of Medical Sciences, Universiti Sains Malaysia , Kelantan , Malaysia
9 Department of Human Physiology, Federal University Dutse , Jigawa , Nigeria
10 Department of Physical Education and Special Motricity, Transilvania University of Brasov , Brasov , Romania
11 Physical Education and Sports, Division of Humanities and Political Sciences, Hellenic Naval Academy , Piraeus , Greece
12 Department of Physical Education and Sport Science, School of Physical Education, Sport Science and Dietetics, University of Thessaly, Karies , Trikala , Greece
13 School of Physical Education and Sport Science, Department of Physical Education and Sport Science , Komotini , Greece
Chen Yung-Sheng
Electronic publication date: 2025 Jun 11
Publication date: 2025
Volume: 13
Electronic Location ID: e19537
Received 2025 Jan 13; Accepted 2025 May 7
Copyright: © 2025 AL-Mhanna et al.
Copyright year: 2025
Copyright holder: AL-Mhanna et al.
License: This is an open access article distributed under the terms of the Creative Commons Attribution License, which permits unrestricted use, distribution, reproduction and adaptation in any medium and for any purpose provided that it is properly attributed. For attribution, the original author(s), title, publication source (PeerJ) and either DOI or URL of the article must be cited.
License URL: https://creativecommons.org/licenses/by/4.0/

Keywords: Aerobic exercise, Resistance training, Overweight, Obesity, Metabolic syndrome, Cardiometabolic health

Funding: The authors received no funding for this work.

==============================
Background

The potential advantages of concurrent aerobic and resistance training (CART) for enhancing cardiometabolic health-related outcomes appear to surpass the outcomes of engaging in aerobic or resistance training alone. The present study aimed to synthesize the available scientific evidence on the effects of CART on body composition, lipid metabolism, and physical function in patients with type 2 diabetes and overweight/obesity.

Methods

PubMed, Scopus, ScienceDirect, Cochrane Library, and Google Scholar were searched from inception to August 7, 2024. The review focused on randomized controlled trials and controlled clinical trials of CART. The Cochrane risk of bias tool was used to assess eligible studies, and the GRADE method to evaluate the reliability of evidence. A random-effects model was used and data were analyzed using standardized mean differences (SMD) and 95% confidence intervals (CI).

Results

A total of 22,878 studies were retrieved; only 20 studies were included, and data were extracted from 1,289 participants (57.0 ± 7.0 years; 31.1 ± 4.6 kg/m2) who met the eligibility criteria. CART group significantly reduced body fat percentage (SMD −0.42, 95% CI [−0.70 to −0.15]), low-density lipoprotein-cholesterol (SMD −0.32, 95% CI [−0.62 to −0.02]), triglycerides (SMD −0.48, 95% CI [−0.71 to −0.24]), total cholesterol (SMD −0.35, 95% CI [−0.58 to −0.12]), and fasting blood glucose levels compared to standard treatment (non-exercising controls). CART significantly increased high-density lipoprotein-cholesterol (SMD 0.44, 95% CI [0.05–0.82]) and improved physical function (cardiorespiratory fitness: SMD 78.78, 95% CI [46.30–111.25]; muscular fitness: SMD 5.19, 95% CI [1.80–8.59]) compared to standard treatment. There were no significant differences in body mass, waist-to-hip ratio, fat mass, and lean body mass between CART and standard treatment. An uncertain risk of bias and poor quality of evidence were observed in the eligible studies.

Conclusions

The present results indicate clear evidence that CART has a beneficial role in the improvement of several cardiometabolic health-related parameters in patients with type 2 diabetes and concomitant overweight/obesity. More trials with robust methodological design are needed to investigate the dose-response effects, training parameters formation, and potential mechanisms.

Introduction

Type 2 diabetes mellitus (T2DM) and obesity are currently major public health concerns globally, significantly increasing the risk of cardiovascular disease morbidity and mortality (Al-Mhanna et al., 2023). As a result, individuals with T2DM and overweight/obesity often experience metabolic and cardiovascular dysfunction due to factors such as sedentary behavior, elevated visceral fat, and disrupted lipid homeostasis (Al-Mhanna et al., 2024a; Bhupathiraju & Hu, 2016). Moreover, these populations often face physical limitations and reduced functional capacity, which negatively impact musculoskeletal health and quality of life (Pataky et al., 2014; Warburton, Gledhill & Quinney, 2001). The economic burden of obesity-related comorbidities is also rising at a concerning rate, posing a significant financial strain on healthcare systems worldwide, and it was estimated to be US $2.0 trillion (Al-Mhanna et al., 2024b; Tremmel et al., 2017). Consequently, investigating cost-effective, non-pharmaceutical interventions has become a key priority for clinicians, practitioners, and public health policymakers, who aim to raise awareness of the essential role of physical exercise in improving community health (Albert et al., 2020).

Physical exercise is recognized as a key element in preventing, managing, and treating individuals who are overweight or obese and have T2DM (Colberg et al., 2016; Kanaley et al., 2022; Kemps et al., 2019; Kim et al., 2019; Umpierre et al., 2011; Zhao et al., 2021). Additionally, exercise has been identified as one of the fastest-growing health and fitness trends globally for people affected by metabolic diseases (Al-Mhanna et al., 2022b; Kercher et al., 2023). According to current international guidelines from the American College of Sports Medicine (ACSM), combining aerobic and resistance training (CART) is highly recommended for improving cardiometabolic health, even without weight loss, in individuals with metabolic health issues (American College of Sports Medicine et al., 2021). Specifically, CART appears to be the most effective training approach for producing positive changes in various cardiometabolic health indicators in adults who are overweight or obese but do not have comorbidities (Al-Mhanna et al., 2024c, 2023; Batrakoulis et al., 2022a). Moreover, both aerobic and/or resistance training can lead to improvements in individuals with T2DM, particularly in terms of glycemic control, cardiovascular health, chronic inflammation, and mental well-being (Al-Mhanna et al., 2024d, 2025; Kanaley et al., 2022; Mendes et al., 2016; Pesta et al., 2017; Qadir et al., 2021; Sabag et al., 2017; Zhao et al., 2021). However, CART has been shown to be more effective than either aerobic or resistance training alone in improving glycemic control in this population (Church et al., 2010; Güngör et al., 2024; Jamka et al., 2022; Sigal et al., 2007), although the optimal CART protocols have yet to be fully established (Kadoglou et al., 2013). Therefore, further research is required, as there is currently no comprehensive scientific evidence regarding the benefits of CART for various health outcomes in T2DM patients with concurrent overweight/obesity.

While it is well-established that physical exercise plays a crucial role in reducing cardiometabolic risk in obesity and T2DM (American College of Sports Medicine et al., 2021), the specific effectiveness of CART in achieving this benefit remains unclear. Therefore, this systematic review and meta-analysis aimed to assess the impact of CART on a wide range of cardiometabolic health parameters in individuals with overweight/obesity and T2DM, including anthropometrics (i.e., body mass and waist-to-hip ratio), body composition (i.e., body fat, fat mass, and lean body mass), lipid metabolism (i.e., high-density lipoprotein cholesterol (HDL-C), low-density lipoprotein cholesterol (LDL-C), total cholesterol (TC), and triglycerides (TG)), glucose metabolism (fasting blood glucose (FBG)), and physical function.

Methods

The methodology employed in the present systematic review and meta-analysis was previously delineated (Al-Mhanna et al., 2023), following the guidelines outlined in the Preferred Reporting Items for Systematic Reviews and Meta-Analyses (PRISMA) statement (Page et al., 2021).

Registration

The study protocol was registered on the Open Science Framework (https://osf.io/evt3w/?view_only=e4b33d370929417ab12c8a3a9e4e9d03).

Literature search strategy

Following a systematic electronic search, articles were gathered from six databases (e.g., PubMed, Scopus, Web of Science, Cochrane Library, Science Direct, and Google Scholar). Four authors (S.B.A.M., A.B.D., H.A., and M.G.) used a combination of keywords and Boolean operators, specifically “OR” and “AND,” to conduct the search until August 7, 2024. The keywords used were (“Diabetes”) AND (“Exercise” OR “Training”), as outlined in Table S1, to retrieve relevant articles. The search strategy was based on keywords related to the PICOS framework (P) Population: T2DM patients with overweight or obesity; (I) Intervention: CART; (C) Comparator: other forms of exercise, non-exercising controls, or standard treatment without exercise; (O) Outcomes: body mass, waist-to-hip ratio, body fat, fat mass, lean body mass, HDL-C, LDL-C, TC, TG, FBG, and physical function; and (S) Study type: randomized controlled trials (RCTs) and controlled clinical trials. A comprehensive examination of the reference lists of the included articles, as well as those of all relevant systematic reviews, was conducted to identify additional articles that met the established inclusion criteria.

Eligibility criteria

The following criteria were used to determine the inclusion of studies in the present analysis: Firstly, the participants were patients diagnosed with T2DM, exhibiting a BMI ranging from 25 to 29.9 kg/m², indicating a state of pre-obesity, or an BMI of 30 kg/m² or higher, categorically designated as obesity. The studies were required to have no specified age limit for participants. The intervention in the studies was required to be CART. The studies were required to investigate at least one of the following primary outcomes in humans: anthropometry (body mass and waist-to-hip ratio), body composition (body fat, fat mass, and lean body mass), lipid metabolism (HDL-C, LDL-C, TC, and TG), and glucose metabolism (FBG). Physical function, as measured by the 6-min walk test and the 30-s sit-to-stand test, was also included as a secondary outcome due to its connection with various cardiometabolic health-related markers. The following criteria were applied to the selection of articles: (v) full-text access was provided, and the articles were published in a peer-reviewed journal from inception until August 7, 2024; (vi) no language restrictions were imposed; and (vii) the studies were RCTs or controlled clinical trials. The following articles were excluded from the analysis: (i) articles that did not assess the relevant outcome measures; (ii) studies involving acute exercise interventions; and (iii) review articles, case reports, studies without a control group, or those with unclear or ambiguous data.

Study selection

Four authors (S.B.A.M., A.B.D., H.A., and M.M.) employed a linear evaluation approach to assess the eligibility criteria. A meticulous examination of the titles, abstracts, and full texts (in instances of uncertainty) was conducted, followed by a thorough evaluation of the remaining articles based on the qualifying criteria. Ultimately, a decision was arrived at. In instances of disagreement or uncertainty, a fifth author (W.S.W.G.) provided assistance by independently implementing the same method. The literature search records were meticulously organized using literature management software (EndNote X9; Clarivate Analytics, Philadelphia, PA, USA).

Data extraction

Two authors (S.B.A.M. and A.B.D.) independently reviewed the full texts of the relevant studies to sample and extract data. The included studies provided substantial data, which was collected and documented, including the first author, publication year, population, gender, sample size, description of the exercise intervention, study length, and outcomes.

Risk of bias assessment

Two authors (S.B.A.M. and A.B.D.) independently evaluated the risk of bias in individual studies following the Cochrane Handbook for Systematic Reviews of Interventions (Higgins et al., 2019). The overall risk of bias for each eligible study was determined according to the following factors: First, the random sequence generation was assessed. Second, the allocation concealment was evaluated. Third, the blinding of participants and personnel was considered. Fourth, the blinding of outcome assessors was examined. Fifth, the completeness of outcome data was assessed. Sixth, the selectivity of outcome reporting was evaluated. Finally, other biases, as described in the Cochrane Handbook for Systematic Reviews of Interventions (Table S2), were considered. The categorization of eligible studies was conducted into three risk levels (high, some concerns, and low) based on the number of factors exhibiting high, unclear, or low risk of bias.

Data analysis

All analyses were conducted using Review Manager 5.4 software (Cochrane Collaboration, https://revman.cochrane.org/info). A random-effects model was employed to present the outcomes, and Cochran’s Q test along with the I² test were used to evaluate heterogeneity. In instances where I² exceeded 50%, a fixed-effects model was implemented to calculate the pooled results, and a subgroup analysis was conducted. The effect size was determined using mean differences (MD) or standardized mean differences (SMD) with 95% confidence intervals (CI). A two-sided p-value of less than 0.05 was considered statistically significant. The GRADEpro methodology (https://www.gradepro.org) was employed to assess the reliability of the evidence, classifying studies as low-, moderate-, or poor-quality evidence (Table S3).

Results

Literature search and selection

From the specified databases, PubMed, Web of Science, Scopus, Science Direct, Cochrane Library, and Google Scholar (Fig. 1), a total of 22,878 studies were obtained. After removing duplicate articles, the number of studies eligible for further evaluation was reduced to 21,195. Through a review of the titles and abstracts based on predetermined inclusion and exclusion criteria, 21,159 studies were excluded. Subsequently, the full text of the remaining 36 articles was carefully examined, excluding 14 articles with reasons. However, two records were subsequent studies of eligible trials included in the present review. Thus, a total of 20 studies were finally included in this review, and data were extracted from 1,289 patients (age: 57.0 ± 7.0; BMI: 31.1 ± 4.6 kg/m2) who met the eligibility criteria (Fig. 1 and Tables S4 and S5).

Figure 1 PRISMA flowchart for search strategy.

The process of study selection for the systematic review and meta-analysis, including identification, screening, eligibility, and inclusion phases. A total of 22,878 records were identified through database searching, and 1,683 duplicates were removed. After screening 21,195 records, 36 full-text articles were assessed for eligibility. Ultimately, 20 studies (22 reports) met the inclusion criteria, while 14 were excluded for reasons such as lack of control group, irrelevant outcomes, or combined interventions.

Literature characteristics

Fourteen out of the 20 studies were from high-income countries (Annibalini et al., 2017; Cuff et al., 2003; Dunstan et al., 1998; Ferrer-García et al., 2011; Gibbs et al., 2012; Hale et al., 2022; Lambers et al., 2008; Loimaala et al., 2003; Magalhães et al., 2019; Maiorana et al., 2002; Scheer et al., 2020; Sigal et al., 2007; Swift et al., 2012; Tessier et al., 2000), two studies, were from upper-middle-income countries (Jorge et al., 2011; Tan, Li & Wang, 2012), and four studies, were from lower-middle-income countries (Motahari Rad et al., 2023; Sabouri et al., 2021; Yavari et al., 2012; Zarei et al., 2021). Only one out of the 20 studies was conducted in a non-clinical setting (home-based intervention). Nine studies conducted short-term exercise interventions lasting 8–12 weeks and 11 studies conducted long-term exercise interventions lasting 16–60 weeks (Table 1).

Table 1 Characteristics of included trials.

The table summarizes key details such as participants’ demographics (age, BMI), comorbidities, study design, recruitment methods, intervention types (aerobic, resistance, or combined), duration, and outcome measures assessed across the included studies.

Reference	Participants’ age (yrs), BMI (kg/m2), population	Comorbidities	Study design	Recruitment and grouping	Control intervention	Intervention group	Duration of intervention	Outcome measures	Pro-instrument measures	
1. Swift et al. (2012)	55.8 ± 8.7,	Cancer,	RCT	Over media, mailers, and community events	No exercise	10 kcal/kg per week expending aerobic and twice a week resistance training.	36 weeks	1. Fasting glucose

2. Body fat

3. Body mass

	1. Immunoassay Siemens, Deerfield 2000

2. –

3. on a GSE 450 electronic scale (GSE Scale Systems, Novi, MI, USA)

	
35.8 ± 6.2,	Neuropathy,	United States	N = 96:	Supervised	
Inter-racial sedentary men and women	Myocardial infarction,	CO = 37:	
Heart catheterization,	EX = 59	
And coronary artery bypass surgery	
2. Ferrer-García et al. (2011)	66.7 ± 8.0,	—	RCT	By clinical interview	Dietary and exercise counseling	45 min of moderate physical (particularly aerobic) for 5 days + Strength training at least 2 days/week	24 weeks	1. Lipid profile

2. Body mass

	1. –

2. In (kg)

	
31.3 ± 6.2,	Spain	N = 84: CO = 40: Ex = 44		Home-based physical education program (HPEP) (verbally and in writing recommendation)	
3. Maiorana et al. (2002)	52.0 ± 2.0,		RCT	Hospital recruitment	They were instructed not to undertake any formal exercise or change in habitual physical activity levels during the study	1 hour of circuit training (a combination of cycle ergometry, treadmill walking, and resistance training)	8 weeks	1. Body mass

2. Body fat

3. Waist-to-hip ratio

4. Plasma lipid profile

5. Plasma fasting glucose

	1.

2. Anthropometric steel tape (Lufkin)

3. Hitachi 917 analyzer (Tokyo, Japan)

	
	29.6 ± 3.4		Australia	N = 16: Ex = 8: CO = 8		Supervised by an experienced exercise physiologist.				
4. Lambers et al. (2008)	55.8 ± 9.7,		RCT	Hospital outpatients	Continued with their normal
daily activities	40 sessions of circuit training (walking or jogging, elbow flexion, and extension, cycling, knee flexion and extension, stepping, cooling down
+ stack weight strengthening	12 weeks	1. Physical function

2. Body mass

3. Lipid profile

	1. 6-min walk test (6MWT), sit-to-stand test

2. Digital balance, stadiometer

3. HDL-C (PEG þ cholesterol oxidase; Roche Diagnostics), triglycerides (glycerol phosphate PAP; Roche Diagnostics), total cholesterol (cholesterol oxidase-PAP; Roche Diagnostics)

	
	28.9 ± 2.8,		Belgium	N = 35: CO = 16: Ex = 19		Exercises				
	From two General Practice centres in the Netherlands and Belgium					Supervised				
5. Hale et al. (2022)	62.7 ± 15.3,		Randomized, two-arm, parallel, open-label trial	via general practices (GP), Diabetes NZ, public media advertising, and health agencies that work with Maori and Pacific communities	Usual care	Diabetes Community Exercise Programme (DCEP) twice weekly. It consists of an aerobic exercise warm-up (5 min), an aerobic and resistance exercise circuit with a focus on major muscle groups (30 min), and flexibility exercises (5 min).	12 weeks	1. Body mass

		
	N/A,		New Zealand	N = 169: CO = 84: Ex = 85		Supervised				
	New Zealand communities									
6. Loimaala et al. (2003)	53.3 ± 5.1,	Hypertension	RCT	Through newspaper advertisement	Conventional treatment only	Conventional treatment together + heart rate-controlled endurance training twice a week + muscle strength training twice a week	60 weeks	1. Physical function

	1. Sit to stand test	
	29.3 ± 3.7		Finland	N = 50: CO = 25: Ex = 24		Supervised				
7. Tessier et al. (2000)	69.3 ± 4.2,		RCT	Hospital outpatients	Continue with their usual activity regimen	Three times a week of rapid walking, strengthening, and stretching exercise	16 weeks	1. Body mass

	1.	
	N/A		Canada	N = 39: CO = 20: Ex = 19		Supervised				
8. Dunstan et al. (1998)	50.3 ± 2.0,		RCT	Non-regular vigorous exercising NIDDM volunteers	No formal exercise	60 min circuit weight training (CWT)	8 weeks	1. Fasting serum glucose

2. Waist-to-hip ratιο

	1. Automated analyzer (Bayer Diagnostics, Sydney, NSW, Australia)

Radioimmunoassay using Tosoh analyzer (Tosoh, Kyobashi, Chuo-ku, Tokyo, Japan)

2. Waist-hip ratio was calculated

from the umbilicus and hip circumference measurement sites

	
	28.3 ± 0.8		Australia	N = 27: CO = 12: Ex = 15		3 days a week
Supervised				
9. Jorge et al. (2011)	57.9 ± 9.8,		RCT	From Diabetes Outpatient Clinic	Continued with their normal
daily activities	3 days per week of cycling at the heart rate corresponding to the lactate threshold (30 min) + seven circuit training as follows: leg press, bench press, lat pull down, seated rowing, shoulder press, abdominal curls, and knee curls (30 min).	12 week	1. Waist-to-hip ratio

2. Cholesterol and triglyceride

	1. By measuring the waist circumference at the narrowest region between the costal margin and iliac crest and then dividing this measurement by the hip circumference measured at its greatest gluteal protuberance

2. Colorimetric methods using commercial kits (Abbott, Abbott Park, IL)

	
	31.2 ± 3.9		Brazil	N = 48: CO =12: Ex = 12					
10. de Oliveira et al. (2012)	57.9 ± 9.8,							3. Body mass

4. Body fat

5. Fasting blood glucose, LDL-C, and HDL-C

	3. in Kg

4. By measuring the waist circumference at the costal margin and iliac crest, dividing by the hip circumference at its greatest gluteal protuberance

5. Colorimetric methods using commercial kits (Abbott, Abbott Park, IL)

	
	31.2 ± 3.9								
11. Sigal et al. (2007)	53.5 ± 7.3,		RCT	Recruited through advertising, physicians, and word of mouth	Maintained their lifestyle	3 times weekly progressive treadmills or bicycle ergometers exercised + seven progressing weight machines
resistant exercises
Supervised	22 weeks	1. Plasma lipid

2. Body composition

3. Body mass

	1. Enzymatic methods on a Beckman-Coulter LX20 analyzer

2. bioelectrical impedance analyser, computed tomography (CT).

	
	35.0 ± 9.6		Canada.	N = 251: CO = 62: Ex = 64						
12. Gibbs et al. (2012)	58.0 ± 5.0,	Hypertension	RCT
United States	Recruited via newspaper advertisements from greater Baltimore city	No further intervention	Three times per week of 10–15 min warm-up, 45 min of aerobic exercise at 60–90% maximum heart rate and a cool down + seven weight training exercises of latissimus dorsi pull down, leg extension, leg curl, bench press, leg press, shoulder press, and seated mid-rowing as two sets of 12–15 repetitions at 50% of 1-repetition maximum.	24 weeks	1. Body fat

2. Lipid profile

	1. Dual X-ray absorptiometry, magnetic resonance imaging

2. Lipids (Cholestech Corp.), insulin (Linco Research Inc.), glucose (Beckman Diagnostics), and HbA1c (Med. Computer Systems).

	
	32.3 ± 5.3,			N = 140: CO = 70: Ex = 70		Supervised				
	Multi-racial									
13. Yavari et al. (2012)	50.9 ± 9.8,		RCT	Clinic DM out-patients	Maintained their lifestyle	Three times per week of a warm-up stage, they worked for 20–30 min on a treadmill or bicycle plus two sets each of eight exercises with 8–10 repetitions on weight machines.
Supervised	52 weeks	1. Blood glucose, total cholesterol, HDL-c, and triglyceride

2. Body fat

	1. Photometric End Point method using Pars Azmoon® enzyme kits (made in Iran)

2. Body composition monitor (model BF500, OMRON®, 2007)

	
	28.8 ± 5.4		Iran	N = 80: CO = 20: Ex=20						
14. Cuff et al. (2003)	63.4 ± 2.2	Obese postmenopausal women	RCT	Hospital recruitment		Warm-up, an aerobic phase, a resistance training phase, and a cooldown to
total class time of 75 min.
Supervised	16 weeks	1. Serum total cholesterol, LDL, apolipoprotein B and HDL cholesterol

2. Abdominal adipose tissue & mid-thigh skeletal

	1. Enzymatic colorimetric test

2. Computed tomography scans

	
	33.3 ± 1.5		Canada	N = 28: CO = 9: Ex = 10						
15. Scheer et al. (2020)	60.9 ± 9.6		Australia
Control Trial	Recruited from the community using local media advertising	Maintained usual activities	Three times a week in a heated community pool of eight aerobic stations alternating with eight resistance stations
supervised	8 week	1. Waist-to-hip ratio

2. Plasma glucose, total cholesterol, HDL, LDL and triglycerides

	1. Triplicate measures using an anthropometric steel tape (Lufkin, Missouri City, TX, USA)

2. –

	
	35.3 ± 6.8			N = 35: CO = 21: Ex = 14						
16. Tan, Li & Wang (2012)	65.9 ± 4.2,		RCT	Recruited via local medical practitioners.	Maintain their usual physical activity habits	Three sessions per week of warm-up period (30 min), moderate aerobic exercise, resistance training (10 min) with five leg muscle exercises (two sets of 10–12 repetitions) and a cool-down
Supervised	24 week	1. Body composition

2. Body mass and

3. Waist-to-hip ratio

4. Blood glucose

5. Lipid profile

	1. GE Prodigy direct digital DEXA bone densitometry (GE Healthcare, Chicago, IL, USA)

2. Kgm−2, Waist girth was measured at the level of the umbilicus horizontally without clothing, while the hip girth at the level of the greatest protrusion of the gluteal muscles with underwear

3. Using Glucose analyzer (YS2300; Yellow Springs, OH, USA)

4. By Cobas Integra Bio-analyzer (Roche, USA) using the standard kits

	
	N/A		China	N = 30: CO = 12: Ex = 18						
17. Zarei et al. (2021)	48.7 ± 10.1,		RCT	N = 26: CO = 13: Ex = 13	Received no intervention	Three sessions per week of aerobic exercise (walking or running) + weight training
Supervised	12 weks	1. Body composition

2. Glucose, lipid profile

	1. Body composition analyzer (InBody 270; South Korea)

2. Enzymatic assay kits

	
	26.7 ± 4.0		Iran							
18. Magalhães et al. (2019)	59.7 ± 6.5,	Hypertension	RCT	Using media advertisements and e-mail	No intervention	Aerobic exercise on cycling at 40–60% of
the heart rate reserve (HRR) + RT included 1 set of 10–12 repetitions.	1 year	1. Body fat

2. Lean body mass

2. Lipid profile

3. Body mass

	1. Dual-energy X-ray absorptiometry (Hologic Explorer W, Waltham, USA)

2. Colored enzymatic tests in an automated

analyzer (auto analyzer Olympus AU640, Beckman Coulter).

3. Electronic

scale with a stadiometer (Seca, Hamburg, Germany).

	
	31.0 ± 5.1		Portugal	N = 38: CO = 22: EX = 16						
19. Magalhães et al. (2020)				CO = 27: EX = 28				4. Fasting glucose

	4. Auto analyzer Olympus AU640, Beckman Coulter

	
20. Annibalini et al. (2017)	60.0 ± 6.8,	–	RCT	---	Usual care
(No intervention)	Aerobic exercise performed on a treadmill with (40% to 65% of heart rate (HR) reserve) and duration (30 to 60 min) + RT gradually increased from 2 to 4 sets of 20 to 12 repetitions from 40% to 60% of 1-repetition maximum (1-RM) for 3 times per week
Supervised	16 weeks	1. Body mass

2. Body composition

3. Fasting glucose

4. Lipid profile

	1. –

2. Dual-energy X-ray absorptiometry (DXA LUNAR® GE Healthcare, Milan, Italy)

3. Routine laboratory

methods in the local clinic by an auto-analyzer (Beckman

Coulter, Milan, Italy).

4. –

	
	29.0 ± 3.8		Italy	N = 16: EX = 8: CO = 8						
21. Sabouri et al. (2021)	52.5 ± 4.8,	–	RCT	N = 28: EX = 15: CO = 13	Subjects in the CON group were asked to retain their habitual physical activity without participating in any exercise program throughout the study.	Aerobic exercise on cycle ergometers + RT for 70 min was performed three training sessions/week
Supervised	12 weeks	1. Body mass

2. Lipid profile

3. HbA1c

	1. Body composition analyzer (InBody 570, Korea)

2. An automated

analyzer (CobasC111; Roche Diagnostics, Indianapolis,

IN, USA)

3. Anion exchange chromatography.

4. East Bio-Pharm,

	
	26.6 ± 2.3		Iran							
22. Kanaley et al. (2022)	43.9 ± 2.5	–	RCT	N = 34: EX = 17: CO = 17	No exercise	Instructed to maintain their current lifestyles.
The RT was 40–80% of 1RM with 15–18 reps. The AE was performed at 75–95% heart rate maximum (HRmax) for 3/week.	12 weeks	1. Body mass

2. Body fat

	1. In (kg)

2. Body composition monitor (model BF500; OMRON®, 2007)

	
	29.6 ± 1.7		Iran							
Note:

EX, exercise; CO, control.

Risk of bias assessment results

The summary of the risk of bias assessment is shown in Fig. 2. Details of the risk of bias judgment per domain for each study are provided in Fig. 3. Specifically, the large majority of eligible studies showed concerns about the randomization, allocation concealment, blinding of participants/personnel and outcome assessment. On the other side, the majority of studies demonstrated a low risk of bias in missing outcome data and a selective reporting process.

Figure 2 Summary of the risk of bias assessment.

Summary of risk of bias across all included studies based on the Cochrane risk-of-bias tool. The figure presents the proportion of studies showing low (green), unclear (yellow), and high (red) risk of bias for each domain assessed.

Figure 3 Risk of bias assessment results.

Each study was evaluated across seven domains: random sequence generation, allocation concealment, blinding of participants and personnel, blinding of outcome assessment, incomplete outcome data, selective reporting, and other sources of bias. Green (+) indicates low risk of bias, yellow (?) indicates unclear risk, and red (−) indicates high risk. Most studies demonstrated low or unclear risk in most domains, with a few showing high risk, particularly in performance bias and allocation concealment (Annibalini et al., 2017; Cuff et al., 2003; Dunstan et al., 1998; Ferrer-García et al., 2011; Gibbs et al., 2012; Hale et al., 2022; Jorge et al., 2011; Lambers et al., 2008; Loimaala et al., 2003; Magalhães et al., 2020, 2019; Maiorana et al., 2002; Motahari Rad et al., 2023; de Oliveira et al., 2012; Sabouri et al., 2021; Scheer et al., 2020; Sigal et al., 2007; Swift et al., 2012; Tan, Li & Wang, 2012; Tessier et al., 2000; Yavari et al., 2012; Zarei et al., 2021).

Primary outcomes

Anthropometry

Body mass was reported in 15 trials involving 795 participants, showing no differences between CART and ST (SMD −0.05, 95% CI [−0.66 to 0.56]; I² = 93%; p = 0.94) following either short- (≤16 weeks) or long-term (>16 weeks) interventions (Fig. 4 and Table S3). The waist-to-hip ratio was assessed in six trials (n = 152), demonstrating no meaningful change between CART and ST (SMD −0.26, 95% CI [−0.69 to 0.17]; I² = 41%; p = 0.24) (Fig. 5 and Table S3).

Figure 4 The effect of CART on body mass.

The forest plot displays standardized mean differences (SMD) and 95% confidence intervals (CI) across studies, with subgroup analysis based on intervention duration (≤16 weeks and >16 weeks). No statistically significant reduction in body mass was observed (SMD = −0.05, 95% CI [−0.66 to 0.56]; p = 0.87), and the overall heterogeneity was high (I² = 93%) (Annibalini et al., 2017; Cuff et al., 2003; Ferrer-García et al., 2011; Hale et al., 2022; Jorge et al., 2011; Lambers et al., 2008; Loimaala et al., 2003; Magalhães et al., 2019; Maiorana et al., 2002; Motahari Rad et al., 2023; Sabouri et al., 2021; Sigal et al., 2007; Swift et al., 2012; Tan, Li & Wang, 2012; Tessier et al., 2000; Zarei et al., 2021).

Figure 5 The effect of CART on the waist-to-hip ratio.

Forest plot illustrating the effect of combined aerobic and resistance training (CART) on waist-to-hip ratio in individuals with type 2 diabetes and overweight/obesity. No significant effect was observed compared to standard treatment (SMD = −0.26, 95% CI [−0.69 to 0.17], p = 0.24), with moderate heterogeneity (I² = 41%) (Dunstan et al., 1998; Jorge et al., 2011; Maiorana et al., 2002; Scheer et al., 2020; Tan, Li & Wang, 2012; Zarei et al., 2021).

Body composition

Body fat was measured in ten trials (n = 551). CART demonstrated a significant reduction in body fat (SMD −0.42, 95% CI [−0.70 to −0.15]; I² = 56%; p = 0.002) compared to standard treatment, showing low-quality evidence (Fig. 6 and Table S3). Fat mass and lean body mass were included in three trials, involving 168 and 179 participants, respectively. No differences were found in both fat mass (SMD −0.19, 95% CI [−0.50 to 0.11]; I² = 0%; p = 0.22) and lean body mass (SMD −0.02, 95% CI [−0.33 to 0.30]; I² = 0%; p = 0.92) between CART and standard treatment (Figs. 7 and 8, Table S3).

Figure 6 The effect of CART on body fat.

The forest plot presents standardized mean differences (SMD) with 95% confidence intervals (CI) for each study and the overall pooled estimate. A statistically significant reduction in body fat was observed in the CART group compared to standard treatment (SMD = −0.42, 95% CI [−0.70 to −0.15]; p = 0.002), with moderate heterogeneity (I² = 56%) (Gibbs et al., 2012; Jorge et al., 2011; Magalhães et al., 2019; Maiorana et al., 2002; Motahari Rad et al., 2023; Sigal et al., 2007; Swift et al., 2012; Tan, Li & Wang, 2012; Yavari et al., 2012; Zarei et al., 2021).

Figure 7 The effect of CART on fat mass.

The forest plot shows standardized mean differences (SMD) with 95% confidence intervals (CI). No significant reduction in fat mass was observed following CART compared to standard treatment (SMD = −0.19, 95% CI [−0.50 to 0.11]; p = 0.22), with low heterogeneity across studies (I² = 0%) (Annibalini et al., 2017; Sigal et al., 2007; Zarei et al., 2021).

Figure 8 The effect of CART on lean body mass.

The forest plot shows standardized mean differences (SMD) with 95% confidence intervals (CI). No significant effect of CART on lean body mass was observed compared to standard treatment (SMD = −0.02, 95% CI [−0.33 to 0.30]; p = 0.92), with no observed heterogeneity (I² = 0%) (Magalhães et al., 2019; Sigal et al., 2007; Zarei et al., 2021).

Lipid metabolism

HDL-C, LDL-C, TC and TG were reported in 11 trials, involving 576, 589, 376 and 626 participants, respectively. CART induced a meaningful increase in HDL-C (SMD 0.44, 95% CI [0.05–0.82]; I² = 76%; p = 0.03) compared to standard treatment, showing very low-quality evidence (Fig. 9, Table S3). CART exhibited significant reductions in LDL-C (SMD −0.32, 95% CI [−0.62 to −0.02]; I² = 63%; p = 0.03), TC (SMD −0.35, 95% CI [−0.58 to −0.12]; I² = 16%; p = 0.003), and TG (SMD −0.48, 95% CI [−0.71 to −0.24]; I² = 44%; p < 0.001) with low-quality evidence between CART and standard treatment (Figs. 10–12, Table S3).

Figure 9 The effect of CART on HDL-C.

The forest plot illustrates a significant improvement in HDL-C following CART compared to standard treatment (SMD = 0.44, 95% CI [0.05–0.82]; p = 0.03), despite moderate heterogeneity among studies (I² = 76%) (Annibalini et al., 2017; Ferrer-García et al., 2011; Gibbs et al., 2012; Jorge et al., 2011; Lambers et al., 2008; Magalhães et al., 2019; Maiorana et al., 2002; Sabouri et al., 2021; Sigal et al., 2007; Tan, Li & Wang, 2012; Zarei et al., 2021).

Figure 10 The effect of CART on LDL-C.

The forest plot shows a statistically significant reduction in LDL-C levels following CART compared to standard treatment (SMD = −0.32, 95% CI [−0.62 to −0.02]; p = 0.03), with moderate heterogeneity observed among studies (I² = 63%) (Annibalini et al., 2017; Ferrer-García et al., 2011; Gibbs et al., 2012; Jorge et al., 2011; Magalhães et al., 2019; Sabouri et al., 2021; Scheer et al., 2020; Sigal et al., 2007; Tan, Li & Wang, 2012; Yavari et al., 2012; Zarei et al., 2021).

Figure 11 The effect of CART on TC.

The forest plot demonstrates a significant reduction in TC following CART compared to standard treatment (SMD = −0.35, 95% CI [−0.58 to −0.12]; p = 0.003), with low heterogeneity (I² = 16%) indicating consistent findings across studies (Annibalini et al., 2017; Ferrer-García et al., 2011; Jorge et al., 2011; Lambers et al., 2008; Magalhães et al., 2019; Maiorana et al., 2002; Sabouri et al., 2021; Scheer et al., 2020; Tan, Li & Wang, 2012; Yavari et al., 2012; Zarei et al., 2021).

Figure 12 The effect of CART on TG.

The meta-analysis revealed a significant reduction in TG following CART compared to standard treatment (SMD = −0.48, 95% CI [−0.71 to −0.24]; p < 0.0001), with moderate heterogeneity (I² = 44%) (Ferrer-García et al., 2011; Gibbs et al., 2012; Jorge et al., 2011; Lambers et al., 2008; Magalhães et al., 2019; Maiorana et al., 2002; Sabouri et al., 2021; Scheer et al., 2020; Sigal et al., 2007; Tan, Li & Wang, 2012; Yavari et al., 2012; Zarei et al., 2021).

Glucose metabolism

FBG reported nine trials involving 387 participants and showing high-quality evidence. CART demonstrated a reduction in FBG (SMD −0.40, 95% CI [−0.70 to −0.10]; I² statistic = 47%; p = 0.009) (Fig. 13 and Table S3) compared to standard treatment.

Figure 13 The effect of CART on FBG.

The meta-analysis showed a significant reduction in FBG following CART compared to standard treatment (SMD = −0.40, 95% CI [−0.70 to −0.10]; p = 0.009), with moderate heterogeneity (I² = 47%) (Annibalini et al., 2017; Dunstan et al., 1998; Ferrer-García et al., 2011; Jorge et al., 2011; Magalhães et al., 2019; Sabouri et al., 2021; Swift et al., 2012; Tan, Li & Wang, 2012; Yavari et al., 2012).

Secondary outcomes

Physical function

6-min walk test

The 6-min walk test was reported in two trials (n = 60). CART exerted a meaningful increase in cardiorespiratory fitness (SMD 78.78, 95% CI [46.30–111.25]; I² statistic = 0%; p < 0.001 compared to standard treatment, demonstrating moderate-quality evidence (Fig. 14 and Table S3).

Figure 14 The effect of CART on cardiorespiratory fitness.

The meta-analysis revealed a significant improvement in cardiorespiratory fitness in the CART group compared to standard treatment (Mean Difference = 78.78, 95% CI [46.30–111.25]; p < 0.00001), with no observed heterogeneity (I² = 0%) (Lambers et al., 2008; Tan, Li & Wang, 2012).

30-s sit to stand test

The 30-s sit-to-stand test was used in two trials (n = 84). CART exhibited a significant improvement in lower body muscular strength (SMD 5.19, 95% CI [1.80–8.59]; I² statistic = 59%; p = 0.003) compared to standard treatment, showing low-quality evidence (Fig. 15 and Table S3).

Figure 15 The effect of CART on lower body muscular strength.

A significant increase in lower body strength was observed in the CART group compared to standard treatment (Mean Difference = 5.19, 95% CI [1.80–8.59]; p = 0.003), with moderate heterogeneity (I² = 58%) (Lambers et al., 2008; Loimaala et al., 2003).

Publication bias

Despite including 22 reports, we were unable to create a funnel plot to detect bias or heterogeneity for specific outcomes due to insufficient studies (less than 10 studies of varying sizes) contributing to each outcome (Higgins, 2008). Therefore, a funnel plot using random effects was generated to explore potential publication bias in studies assessing HDL-C, LDL-C, TG, TC, BMI, and body fat. The number of studies for each outcome varied (with 10 or more studies of various sizes) (Higgins, 2008). The funnel plot illustrating the impact of exercise on HDL-C, LDL-C, TG, TC, BMI, and body fat shows a symmetrical distribution on both sides, indicating minimal publication bias in the results, as depicted in Figs. S1–S6.

Discussion

Our study aimed to evaluate the influence of CART on a wide-ranging range of cardiometabolic health-related indicators in individuals with overweight/obesity and T2DM. The main findings show that CART leads to favorable changes in body composition, lipid and glucose metabolism, and physical function in patients with T2DM and concurrent overweight/obesity. While aerobic and resistance training have been recognized as effective exercise strategies for improving cardiovascular disease risk factors in people with T2DM (Grace et al., 2017; Kelley & Kelley, 2007; Nery et al., 2017; Yang et al., 2014), our results suggest that CART may be the optimal exercise approach for individuals with poor metabolic health due to the combined presence of T2DM and overweight/obesity.

Anthropometry and body composition

Populations with metabolic dysregulation often experience abdominal obesity, which is associated with visceral and ectopic fat, contributing to chronic inflammation that exacerbates insulin resistance (Moore & Shah, 2020). This review suggests that CART improves body fat but does not significantly affect other anthropometric or body composition parameters in individuals with T2DM and overweight/obesity. These findings highlight the need for further investigation into these important variables, as the observed changes remain uncertain. Given that weight management is crucial for individuals with poor metabolic health, our results are particularly relevant for people with T2DM and concomitant overweight/obesity.

CART has been recognized as the most effective form of exercise for improving various anthropometric and body composition measures in individuals with excess weight (Batrakoulis et al., 2018, 2022a; Yapici et al., 2023). Other relevant studies have shown similar effects of CART on anthropometry and body composition in individuals with T2DM and concurrent overweight/obesity (Pan et al., 2018; Zhao et al., 2021). However, further randomized controlled trials (RCTs) are needed to determine whether CART can induce beneficial changes in visceral adiposity, which is linked to higher cardiovascular disease morbidity and mortality risks (Moore & Shah, 2020).

Lipid metabolism

Individuals with T2DM and concurrent overweight/obesity are likely to experience poor lipid profile, promoting cardiovascular complications (Costanzo et al., 2015; Tan et al., 2023). Following the current guidelines published by the American Diabetic Association, it is crucial for these people to maintain normal blood lipid levels, aiming to reduce the risk of developing comorbidities (Colberg et al., 2016). In our review, CART exhibited meaningful improvements in blood lipids. In general, various exercise modes induce favorable alterations in lipid homeostasis in people with T2DM (Kelley & Kelley, 2007; Magalhães et al., 2020), which is an observation aligned with the present results. Such an outcome cannot be explained here; however, the presence of excessive weight in conjunction with T2DM may play some role in the simultaneous control of glucose and lipid metabolism due to a complex, low-grade chronic inflammation. Also, CART appears the best exercise mode for improving lipid profile among individuals with an unhealthy weight (Batrakoulis, 2022a) however, several alternative types of exercise induce positive alterations in blood lipids and lipoproteins among populations with metabolic health impairments and excessive weight (Batrakoulis, 2022a, 2022b, 2022c; Batrakoulis & Fatouros, 2022) promoting time-efficient and positive experiences in this population within real-world gym settings.

Glucose metabolism

Our results indicate that CART significantly reduces FBG in individuals with T2DM and concurrent overweight/obesity. Notably, these reductions in FBG are associated with a decreased risk of developing T2DM-related morbidity and mortality (Moore & Shah, 2020). This finding is consistent with previous meta-analyses that have explored the impact of exercise training on glucose homeostasis in individuals with T2DM, both with and without concurrent overweight/obesity (Zhao et al., 2021). It is important to note that long-term exercise interventions (greater than 12 weeks) appear to be more effective than short-term ones (12 weeks or less), underscoring the enduring benefits of exercise training in glucose control for individuals with T2DM and overweight/obesity (Zou et al., 2016). Moreover, CART has been identified as the most effective exercise modality for improving glycemic management in populations with impaired metabolic health, outperforming other types of exercise like endurance, resistance, or high-intensity interval training (Batrakoulis, Jamurtas & Fatouros, 2021; Batrakoulis et al., 2022a; Zhao et al., 2021). In summary, the positive impact of CART on glucose metabolism may be partly explained by the high prevalence of visceral adiposity in individuals with T2DM and concurrent overweight/obesity (Bays et al., 2008). Consequently, the beneficial changes observed after CART interventions could be linked to the activation of key molecular mechanisms that regulate whole-body glucose homeostasis, particularly those associated with visceral and ectopic fat.

Physical function

Individuals with impaired metabolic health often have low cardiorespiratory fitness and functional capacity, which increases their risk of developing comorbidities (Al-Mhanna et al., 2022a; Colberg et al., 2010; Jensen et al., 2014). In contrast, high cardiorespiratory fitness is linked to a lower risk of cardiovascular disease morbidity and mortality in apparently healthy individuals. This has been identified as a more influential factor for cardiometabolic health than improvements in anthropometrics and body composition (Afolabi et al., 2023; McAuley et al., 2016). Additionally, cardiorespiratory fitness is inversely related to liver fat accumulation and insulin resistance, although no significant differences were found between individuals with T2DM and those without T2DM in terms of cardiorespiratory fitness levels (Sabag et al., 2021). Regarding the positive effects of exercise training on cardiorespiratory fitness, CART has been recognized as the optimal form of exercise for improving aerobic capacity in individuals with overweight/obesity but without comorbidities (Batrakoulis et al., 2022a; O’Donoghue et al., 2021). Notably, exercise protocols combining endurance and resistance training in a single session have proven effective in improving various physical fitness measures in sedentary individuals with excess body weight (Batrakoulis et al., 2018, 2022b, 2021). The significant increase in aerobic capacity may be attributed to the favorable mitochondrial adaptations, increased skeletal muscle capillarization, and enhanced oxidative metabolism induced by CART (Murlasits, Kneffel & Thalib, 2018). However, the findings supporting the positive role of CART in improving cardiorespiratory fitness should be interpreted cautiously, as only two eligible studies were included in the meta-analysis.

With regard to muscular fitness, CART demonstrated a substantial increase in lower body strength. However, the paucity of available data precludes the presentation of compelling evidence regarding the effectiveness of CART on muscular strength and functionality among individuals with T2DM and concurrent overweight/obesity. Nevertheless, this is an important observation given that muscular strength has been documented as a prognosticator of all-cause mortality in adults with no chronic diseases (Carbone et al., 2020). Despite the absence of substantial evidence supporting the pivotal role of muscular fitness in cardiometabolic health among individuals with excessive weight, favorable changes in muscular strength have been associated with a reduced risk of cardiovascular disease in these populations (Carbone et al., 2020). However, CART is in line with the current physical activity guidelines recommending the integration of cardiovascular and neuromuscular stimulus into either a single session or separate sessions weekly (Bull et al., 2020). Interestingly, CART has been reported as a more effective type of exercise for improving musculoskeletal fitness-related parameters compared to aerobic, interval, or resistance training alone (Batrakoulis, 2022a).

The inclusion of the muscle-strengthening component in CART may be the factor that explains the meaningful CART-induced strength increases due to neural adaptations (Strasser, Arvandi & Siebert, 2012), increased muscle fiber activation, mitochondrial biogenesis and glucose transport commonly observed among individuals with metabolic health impairments and excessive weight (Schjerve et al., 2008)

Implications for future research

Although CART appears to be a promising exercise modality for individuals with T2DM and concurrent overweight/obesity, particularly in improving various cardiovascular disease risk factors, there is a lack of strong evidence regarding its effectiveness in real-world settings. Despite recent exercise prescription guidelines for individuals with T2DM (Aschner, 2017; Colberg et al., 2010; Mendes et al., 2016), further studies are needed to determine the optimal training parameters, such as frequency, intensity, and duration. These studies would help guide physicians and exercise professionals in prescribing evidence-based CART programs for individuals with T2DM and overweight/obesity (Batrakoulis, Jamurtas & Fatouros, 2022). Among the 20 eligible RCTs reviewed, the most common CART protocol used either inter- or infra-session formats, typically 2–3 times per week. Circuit-based CART was also widely utilized, combining cardiorespiratory and muscle-strengthening exercises in an efficient manner. Furthermore, additional research should focus on exploring the dose-response relationship between CART and cardiometabolic health parameters, particularly in real-world exercise contexts, as previously suggested (Batrakoulis et al., 2019). Such studies would help optimize the application of CART, determining whether it can serve as the most comprehensive exercise solution for individuals with poor metabolic health in everyday environments.

Limitations

Our meta-analysis has several limitations, and therefore, the results should be interpreted with caution. The included studies displayed inconsistencies in the training parameters used during the interventions, leading to considerable heterogeneity among the studies. Additionally, the present study indicates that beneficial adaptations from CART do not occur in younger individuals, as the mean age of participants was 57 ± 7 years. Therefore, the findings cannot be generalized to younger age groups with T2DM and excess weight. Furthermore, given the outcome measures included, the role of CART in improving cardiometabolic health in this specific population remains unclear, due to the lack of data on factors such as glucose homeostasis, blood pressure, inflammation, and oxidative stress.

Conclusions

This meta-analysis provides valuable evidence on the use of CART for patients with T2DM and concurrent overweight/obesity as a supportive element in the comprehensive management of poor metabolic health. The findings offer clear evidence that CART plays a beneficial role in improving various cardiometabolic health markers, such as anthropometrics, body composition, lipid profile, fasting glucose levels, and physical function in individuals with T2DM and concomitant overweight/obesity. However, further studies with robust methodologies are needed to explore the dose-response relationship, optimal training parameters, and the mechanisms underlying these positive adaptations. This review also emphasizes the need for additional RCTs to assess more comprehensive glucose metabolism outcomes, as well as resting cardiovascular function, inflammation, and oxidative stress markers, to better understand the effects of CART in individuals with T2DM and concurrent overweight/obesity.

Supplemental Information

Supplemental Information 1 Prisma checklist for search strategy.

Supplemental Information 2 Search strategy used for the systematic review.

The databases (e.g., PubMed, Scopus, Google Scholar, Cochrane Library) and the specific search algorithms applied to identify studies related to exercise, training, and type 2 diabetes.

Supplemental Information 3 Risk of bias assessment for the included studies.

The quality and potential biases in each study, categorized into domains such as random sequence generation, allocation concealment, blinding of participants and personnel, blinding of outcome assessment, incomplete outcome data, selective reporting, and other biases. Judgements (e.g., low, unclear, or high risk) are provided with supporting justifications for each domain.

Supplemental Information 4 Summary of quality assessment findings using the GRADE framework.

A certainty assessment of the evidence for various outcomes (e.g., body mass, body fat, lipid profile, fasting blood glucose, and physical function). Certainty levels (very low, low, moderate) are determined based on factors such as risk of bias, inconsistency, indirectness, imprecision, and other considerations, with details on patient numbers and standardized mean differences (SMD) or mean differences (MD) for each outcome.

Supplemental Information 5 Excluded full-text articles with reasons.

The studies excluded from the systematic review after full-text assessment, along with the specific reasons for exclusion, such as unrelated outcome measures, combined diet and exercise interventions, lack of a control group, inclusion of both diabetic and non-diabetic participants, or control groups performing exercise.

Supplemental Information 6 The subsequent reports of original studies.

The additional reports derived from original studies and specifies the reasons for their inclusion in the systematic review, based on the reported parameters such as body fat, body weight, fasting blood glucose, LDL-C, and HDL-C.

Supplemental Information 7 Forest plot of the effects of Concurrent Aerobic and Resistance Training (CART) on Body Mass Index (BMI), based on a meta-analysis of the included studies.

Supplemental Information 8 Forest plot of the effects of CART on body fat.

Forest plot of the effects of Concurrent Aerobic and Resistance Training (CART) on body fat, based on a meta-analysis of the included studies.

Supplemental Information 9 Forest plot of the effects of CART on HDL-C.

Forest plot of the effects of Concurrent Aerobic and Resistance Training (CART) on HDL-C levels, based on a meta-analysis of the included studies.

Supplemental Information 10 Forest plot of the effects of CART on LDL-C.

Forest plot of the effects of Concurrent Aerobic and Resistance Training (CART) on LDL-C levels, based on a meta-analysis of the included studies

Supplemental Information 11 Forest plot of the effects of CART on TG.

Forest plot of the effects of Concurrent Aerobic and Resistance Training (CART) on triglyceride (TG) levels, based on a meta-analysis of the included studies.

Supplemental Information 12 Forest plot of the effects of CART on TC.

Forest plot of the effects of Concurrent Aerobic and Resistance Training (CART) on total cholesterol (TC) levels, based on a meta-analysis of the included studies.

Supplemental Information 13 Overview of the effects of Concurrent Aerobic and Resistance Training (CART) compared to standard treatment without exercise in patients with type 2 diabetes and overweight/obesity.

Based on 20 studies (N = 1,289), CART positively impacted body composition (↓ body fat, ↑ lean body mass), muscular fitness (↑ strength), glycolipid metabolism (↓ fasting blood glucose, ↓ total cholesterol, ↓ LDL-C, ↑ HDL-C, ↓ triglycerides), and cardiorespiratory fitness (↑ 6-min walk test). The average participant age was 57 ± 7 years, with a BMI of 31.1 ± 4.6 kg/m².

Additional Information and Declarations

Competing Interests

The authors declare that they have no competing interests.

Author Contributions

Sameer Badri AL-Mhanna conceived and designed the experiments, prepared figures and/or tables, and approved the final draft.

Abdullah F. Alghannam conceived and designed the experiments, prepared figures and/or tables, and approved the final draft.

Nouf H. Alkhamees conceived and designed the experiments, prepared figures and/or tables, and approved the final draft.

Bodor Bin Sheeha conceived and designed the experiments, prepared figures and/or tables, and approved the final draft.

Norsuhana Omar conceived and designed the experiments, prepared figures and/or tables, and approved the final draft.

Hani Albalawi performed the experiments, authored or reviewed drafts of the article, and approved the final draft.

Mehmet Gülü performed the experiments, authored or reviewed drafts of the article, and approved the final draft.

Umut Canli performed the experiments, authored or reviewed drafts of the article, and approved the final draft.

Hafeez Abiola Afolabi performed the experiments, authored or reviewed drafts of the article, and approved the final draft.

Bishir Daku Abubakar performed the experiments, authored or reviewed drafts of the article, and approved the final draft.

Georgian Badicu analyzed the data, prepared figures and/or tables, and approved the final draft.

Rozaziana Ahmad analyzed the data, prepared figures and/or tables, and approved the final draft.

Gerasimos V. Grivas analyzed the data, prepared figures and/or tables, and approved the final draft.

Alexios Batrakoulis analyzed the data, prepared figures and/or tables, and approved the final draft.

Data Availability

The following information was supplied regarding data availability:

This is a systematic review/meta-analysis.

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
