# Peer review of "Impact of concurrent aerobic and resistance training on body composition, lipid metabolism and physical function in patients with type 2 diabetes and overweight/obesity: a systematic review and meta-analysis"

_PeerJ, doi:10.7717/peerj.19537_

## Round 0.1 · original submission · Major Revisions

Dear Authors

Two experts in the study field have reviewed the manuscript. Your study has been reviewed with comprehensive comments to improve the quality of the manuscript. I would encourage you to highlight the study's novelty as the point raised by the reviewers. We invite you to submit a revised version of the manuscript addressing the reviewers’ comments.

We look forward to receiving your revised manuscript.

Best regards

Yung-Sheng Chen, Ph.D.
Academic Editor

Reviewer 1 ·

Basic reporting

This systematic review and meta-analysis examined the effects of concurrent aerobic and resistance training (CART) on body composition, lipid metabolism, and physical function in patients with type 2 diabetes (T2D) and excess adiposity. The authors suggested that CART significantly reduced body fat percentage, low-density lipoprotein cholesterol, triglycerides, total cholesterol, and fasting blood glucose levels, increased high-density lipoprotein cholesterol, and improved physical function, including cardiorespiratory fitness and muscular fitness. The findings are relevant for clinical practice, but the manuscript needs improving.

1. Since CART is a named exercise approach, why not indicate CART directly in the title, instead of a confusing phrase of concurrent training? The eligibility criteria limited the participants being overweight and obesity, which is not necessary for people with excessive adiposity. Please use proper language in the title.
2. The current title is very long. The outcome information may not be necessary in the title.
3. The RoB2 tool is preferred for this context.
4. ST is a very uncommon abbreviation. Please avoid.
5. For physical exercise training, dose-response analysis is very necessary.
6. In the eligibility criteria, it is important to make statements very clear. For example, did the authors include if the trials included adults with type 2 diabetes and 70% of those overweight or obesity?
7. For most continuous variables, mean difference rather than SMD is preferred unless mean difference is infeasible.
8. Did the authors consider the treatment duration or the effect size after the treatment ended.
9. There are too many Revman forest plots in the study. It is difficult to read. Please move them to the Appendix and add the summary of findings table in the main text.

Experimental design

The study desgin is generally OK, but some improvement is necessary based on the above comments.

Validity of the findings

The major issue of this manuscript is the unclear eligibility criteria.

Additional comments

See above.

·

Basic reporting

No comment

Experimental design

Research question well defined, relevant & meaningful. It is stated how research fills an identified knowledge gap.

Validity of the findings

Conclusions are well stated, linked to original research question & limited to supporting result.

Additional comments

The article: Impact of concurrent training on body composition, lipid metabolism and physical function in patients with type 2 diabetes and excess adiposity: A systematic review and meta-analysis is presented

It is well structured and provides scientific knowledge of the subject. I leave some comments that may enrich it.
Line 52; what are standard treatment (ST)? please explain it.
Could you explain please if you looked for studies with participants with obesity or excess fat, how was this classified?
According this sentence line 70; The economic burden of obesity-related comorbidities is also rising at a concerning rate, posing a significant financial strain on healthcare systems worldwide. Do you have the economic expenditure in dollars? by continent ?
This sentence need a reference, Consequently, investigating cost-effective, non-pharmaceutical interventions has become a key priority for clinicians, practitioners, and public health policymakers, who aim to raise awareness of the essential role of physical exercise in improving community health.
In the objective you need to clarify the variables more deeply, you can add them in parentheses
Therefore, this systematic review and meta-analysis aimed to assess the impact of CART on a wide range of cardiometabolic health parameters in individuals with overweight/obesity and T2DM, including anthropometrics, body composition(i.e., xxx xxx xxx ), lipid metabolism (i.e., xxx xxx xxx ), and physical function.

Add information of the principal variables of studio, not only metabolic profile, add according FBG and cholesterol, etc.
2.3 Eligibility Criteria
You have studies with morbid obesity?? You need de improved this criteria’s(i) participants were
patients with T2DM and concurrent overweight (BMI 25 29.9 kg/m²) or obesity (BMI g30 kg/m²);

2.4 Study Selection
It is ok.
Results are ok
Discussion
Please add the objective to the beginning of the discussion.
Line 275 To the best of our knowledge, our study presents the first evidence on the effectiveness of CART in improving various cardiometabolic health-related parameters. ADD the type of cardiometabolic health-related parameters… (i.e., FBG, HLDL, etc).
This sentence needs to be discus better, comparing with other studies, according body fat (%) mainly.
.1 Anthropometry and Body Composition
285 This review suggests that CART improves body fat but does not significantly affect other
286 anthropometric or body composition parameters in individuals with T2DM and
287 overweight/obesity. These findings highlight the need for further investigation into these important
288 variables, as the observed changes remain uncertain. Given that weight management is crucial for
289 individuals with poor metabolic health, our results are particularly relevant for people with T2DM
290 and concurrent excess weight.
Please add this sense in the correct place.
These populations often experience abdominal obesity, which is associated with visceral and ectopic fat, contributing to chronic inflammation that exacerbates insulin resistance [54].

This sentence is too long please rewritten and add references.
As for muscular fitness, CART showed a substantial increase in lower body strength;
355 however, the limited available data are not able to provide strong evidence of the effectiveness of
356 CART on muscular strength and functionality among people with T2DM and concurrent
357 overweight/obesity. Nevertheless, this is an important observation given that muscular strength
358 has been documented as a prognosticator of all-cause mortality in adults with no chronic diseases
359 [79]. Although there is no robust evidence concerning the vital role of muscular fitness in
360 cardiometabolic health among people with excessive weight, favorable changes in muscular
361 strength are linked to a lower cardiovascular disease risk in these populations [79].

---

## Round 0.2 · accepted · Accept

Dear Authors,

I would like to express my appreciation for your patience and efforts to improve the quality of the manuscript. The reviewer has suggested to add a practical implication regarding the findings of the study. Please consider this suggestion. Your submission is now endorsed for acceptance of publication in PeerJ. Congratulation!!!

Thank you for submitting your article to PeerJ. I look forward to receiving your research and review articles in the future.

Best Regards
Ph.D. Yung-Sheng Chen

·

Basic reporting

I would like to thank the authors for addressing my earlier suggestions, providing additional data, and clarifying their methods. I appreciate their effort. In general the article has improved a lot.

Abstract
OK
Introduction
ok

Materials and methods
ok

Results:
ok

Discussion
Discussing the results according to the duration (weeks) of each intervention in more depth would be relevant.

Implications for Future Research

In relation to this, I would add a practical implication based on your results. How many weeks, for such a variable xxx, what intensity, how many sessions, etc. This is necessary due to the variety of intervention times. This way, to have clinical decisions to intervene and treatment in non-pharmacological way to T2D.

Conclusion
ok

Experimental design

ok

Validity of the findings

ok

Additional comments

ok